# Idiosyncratic Fitness Costs of Ampicillin-Resistant Mutants Derived from a Long-Term Experiment with *Escherichia coli*

**DOI:** 10.3390/antibiotics11030347

**Published:** 2022-03-06

**Authors:** Jalin A. Jordan, Richard E. Lenski, Kyle J. Card

**Affiliations:** 1Department of Chemistry, Michigan State University, East Lansing, MI 48824, USA; jorda105@msu.edu; 2Perelman School of Medicine, University of Pennsylvania, Philadelphia, PA 19104, USA; 3Department of Microbiology and Molecular Genetics, Michigan State University, East Lansing, MI 48824, USA; lenski@msu.edu; 4BEACON Center for the Study of Evolution in Action, Michigan State University, East Lansing, MI 48824, USA; 5Department of Translational Hematology and Oncology Research, Cleveland Clinic Lerner Research Institute, Cleveland, OH 44106, USA

**Keywords:** antimicrobial resistance, fitness costs, pleiotropy, relative fitness, tradeoffs, experimental evolution

## Abstract

Antibiotic resistance is a growing concern that has prompted a renewed focus on drug discovery, stewardship, and evolutionary studies of the patterns and processes that underlie this phenomenon. A resistant strain’s competitive fitness relative to its sensitive counterparts in the absence of drug can impact its spread and persistence in both clinical and community settings. In a prior study, we examined the fitness of tetracycline-resistant clones that evolved from five different *Escherichia coli* genotypes, which had diverged during a long-term evolution experiment. In this study, we build on that work to examine whether ampicillin-resistant mutants are also less fit in the absence of the drug than their sensitive parents, and whether the cost of resistance is constant or variable among independently derived lines. Like the tetracycline-resistant lines, the ampicillin-resistant mutants were often less fit than their sensitive parents, with significant variation in the fitness costs among the mutants. This variation was not associated with the level of resistance conferred by the mutations, nor did it vary across the different parental backgrounds. In our earlier study, some of the variation in fitness costs associated with tetracycline resistance was explained by the effects of different mutations affecting the same cellular pathway and even the same gene. In contrast, the variance among the ampicillin-resistant mutants was associated with different sets of target genes. About half of the resistant clones suffered large fitness deficits, and their mutations impacted major outer-membrane proteins or subunits of RNA polymerases. The other mutants experienced little or no fitness costs and with, one exception, they had mutations affecting other genes and functions. Our findings underscore the importance of comparative studies on the evolution of antibiotic resistance, and they highlight the nuanced processes that shape these phenotypes.

## 1. Introduction

Antibiotic resistance is a topic of growing concern. Since the introduction of penicillin, society has relied on antibiotics to treat many serious bacterial infections. However, a tension exists between the long times required for the discovery and introduction of new drugs to combat pathogens and the rapid evolution and global spread of bacteria resistant to these drugs. This “arms race” has threatened the effectiveness of antibiotics and spurred a renewed emphasis on drug discovery [1], antibiotic stewardship [2], and studies of the evolutionary processes that give rise to resistance [3].

When a bacterium evolves resistance, either by mutation or horizontal gene transfer, it will have a higher fitness than its sensitive counterparts, and it will therefore be favored in an environment containing the antibiotic at sufficient concentration. Nonetheless, resistance often comes at the cost of a reduced growth rate, such that sensitive cells outcompete resistant variants in drug-free environments [4,5,6,7]. A resistant genotype’s relative fitness, in both types of environments, is therefore an important measure for understanding its clinical impact [8]. For example, the fitness effect of a resistance mutation determines how well it spreads during drug therapy, and its rate of disappearance upon cessation of treatment [8,9,10]. However, a strain’s genetic background can also affect the fitness costs of resistance and therefore alter these dynamics [11,12,13,14,15].

In previous papers, we investigated how genetic background affects the phenotypic and genotypic evolution of drug resistance. We subjected clones, isolated from several laboratory-evolved populations of *Escherichia coli*, to one of four antibiotics in a single round of selection. We found that a strain’s genotype sometimes affected both its resistance potential [16] and the mutational paths by which it evolved resistance [17]. We then examined the competitive fitness of the tetracycline-resistant mutants [10]. We found that the resistant mutants grew, on average, ~8% slower than their sensitive counterparts in the absence of the drug, but with significant among-line heterogeneity in these fitness costs. We asked whether this heterogeneity was explained by the level of resistance conferred by the mutations [7] or some other factors. Our results showed that the level of resistance did not explain the variation in fitness costs, nor did the genetic background. Instead, the variation among lines was explained, in part, by different mutations that arose in the same gene, on the same genetic background, and conferred the same phenotypic resistance.

Here, we extend this work to examine the fitness costs of ampicillin-resistant mutants that evolved from the same parental strains. As we saw with tetracycline resistance [10], the ampicillin-resistant mutants are less fit, on average, than their progenitors in a drug-free environment, and with significant heterogeneity in fitness costs. Once again, neither the level of resistance conferred by the mutations, nor the different genetic backgrounds can explain this variation. Instead, the variation in fitness largely reflects different sets of genes in which the resistance mutations occurred, with some targets associated with high costs and others imposing little or no cost.

Our results largely support other studies, in particular that antibiotic resistance is often, but not always, a detriment to growth in environments where resistance is not essential for survival [4,5,6,7]. Nevertheless, there is value in finding and reporting concordant results across different systems and studies, given the growing problem of antibiotic resistance. The present study also highlights some subtle, but important, differences from our earlier work. In particular, the variation in the fitness costs of ampicillin resistance is not explained by different mutations in the same genes, but rather by mutations affecting different targets. This difference in the source of heterogeneity of fitness costs between ampicillin and tetracycline resistance emerged despite using the same parent clones, environment, and experimental protocol in our two studies. We believe this difference illustrates the value of comparative studies on the evolution of resistance.

## 2. Materials and Methods

### 2.1. Bacterial Strains

The long-term evolution experiment, or LTEE, is described in detail elsewhere [18]. In brief, twelve populations of *E. coli* were founded from a common ancestral strain, called REL606. These populations have been propagated for over 30 years and 70,000 bacterial generations by daily 100-fold dilutions in Davis Mingioli medium [19] supplemented with 25 µg/mL glucose (DM25).

We previously inoculated REL606 and clones isolated at 50,000 generations from four populations (denoted Ara−5, Ara−6, Ara+4, and Ara+5) into replicate cultures of permissive Luria Bertani (LB) medium [16,19]. We spread these cultures on a series of agar plates containing two-fold increasing concentrations of ampicillin. We chose ampicillin because it is widely used in both microbiological and evolutionary studies. We quantified each strain’s evolvability, which we defined as the maximum increase in resistance from its initial inhibitory concentration during one round of drug selection [16]. We later sequenced the complete genomes of the resistant mutants that formed colonies at the highest drug concentrations [17].

In this study, we examined the competitive fitness of the mutants. Specifically, we analyzed four mutants evolved from the LTEE ancestor and three mutants from each derived background, for a total of 16 mutants (Appendix A). Strains REL607, REL10948, and REL11638 were used as common competitors. REL607 is a spontaneous Ara^+^ revertant of REL606 [18], REL10948 is an Ara^−^ clone isolated from population Ara−5 at generation 40,000, and REL11638 is a spontaneous Ara^+^ revertant of that clone [20,21]. The Ara marker is selectively neutral in DM25, and it serves to distinguish competitors during fitness assays because Ara^−^ and Ara^+^ cells form red and white colonies, respectively, on tetrazolium-arabinose (TA) indicator agar plates. We used REL607 as the common competitor for REL606 and the four ampicillin-resistant clones evolved from it, and the 40,000-generation clones as common competitors for the four 50,000-generation clones and the twelve mutants that evolved from them.

### 2.2. Fitness Assays

We performed competition assays in the absence of antibiotics to quantify the relative fitness of each ampicillin-resistant mutant and its sensitive parental clone. Fitness was measured under the same conditions as the LTEE, except the medium contained 250 µg/mL glucose (DM250). Each resistant mutant and its sensitive parent competed, in paired assays, against the same common competitor with the opposite Ara-marker state. To perform the fitness assays, competitors were revived from frozen stocks and acclimated to the DM250 medium for two days. We then diluted each competitor 1:200 into fresh medium, and a sample was immediately plated on TA agar [19] to assess their initial frequencies based on colony counts. We then propagated the cultures for three days, with 1:100 dilutions in fresh medium each day. On day three, samples were again plated on TA agar to assess the competitors’ final densities. We calculated relative fitness as the ratio of the realized growth rate of the clone of interest (either a resistant clone or its sensitive parent) to that of the common competitor. The fitness of a resistant mutant was then normalized by dividing it by the fitness of the paired assay using its parental strain. In other words, if the fitness of the mutant relative to the common competitor is *X*, and if the fitness of the sensitive parent relative to the same competitor is *Y*, then we express the fitness of the resistant mutant relative to its parent, *W*, as *X*/*Y*. We tested the fitness of each of the 16 ampicillin-resistant mutants and the 16 sensitive parents relative to their respective common competitors with five-fold replication, for a total of 160 competitions. The relative fitness values were log_e_-transformed before the statistical analyses reported in the Results below. Note that we report the reduction in fitness (i.e., cost) of resistant mutants relative to their sensitive parents as a percentage. This value is typically near, but does not precisely equal, the reported log_e_-transformed fitness. For example, if log_e_ *W* = −0.2, then *W* = exp(−0.2) = 0.8187. The fitness cost (1 − *W*) is thus 0.1813, or ~18%.

We provide the datasets and analysis code for this study on GitHub at https://github.com/KyleCard/LTEE-ampicillin-fitness-costs (accessed on 6 February 2022).

## 3. Results

Ampicillin-resistant mutants have reduced fitness in the absence of antibiotic. We first ask whether the ampicillin-resistant mutants are, on average, less fit than their sensitive parental strains during competition assays in the absence of the drug. The grand mean of the log_e_-transformed fitness values is −0.1108, which means the resistant mutants grow on average ~10% more slowly than their parents. This value deviates significantly from the null hypothesis that the resistant and sensitive strains have equal fitness (*t_s_* = 2.9169, 15 d.f., one-tailed *p* = 0.0053). It is interesting, however, that about half of the resistant mutants show little or no fitness costs relative to their sensitive parents (Figure 1).

### 3.1. Fitness Costs Significantly Vary among Ampicillin-Resistant Mutants

We measured the relative fitness of each resistant mutant with five-fold replication. The variation in fitness among the mutants is far greater than expected from the variation between replicate assays (Table 1). This result shows that measurement noise cannot explain the variation in fitness costs among the 16 ampicillin-resistant mutants (Figure 1).

As outlined in Card et al. [10], there are several plausible explanations for this variation in fitness costs, and they are not mutually exclusive. The costs might scale with the level of resistance conferred by mutations, there could be genetic-background effects, secondary mutations may have hitchhiked with some mutations that confer resistance, and different resistance mutations may have idiosyncratic effects. In terms of idiosyncratic effects, the fitness costs could vary across pathways that confer resistance by different mechanisms, among mutations in different genes in the same pathway, or even between different mutations in the same gene. We examine these possibilities in the following sections.

### 3.2. Level of Resistance Does Not Explain the Variation in Fitness Costs

The resistant mutants vary in both the minimum inhibitory concentration (MIC) achieved after a single round of exposure to ampicillin and the magnitude of the resistance increase relative to their parental strains (Figure 2). For example, some mutants evolved MIC values that are two-, four-, or even eight-fold higher than their progenitors, while two mutants did not achieve even a two-fold increase in resistance based on our earlier study [16]. Mutations that provide greater resistance might be expected to have higher fitness costs [22]. We tested this possibility by examining the correlation between the log_e_-transformed fitness values of the 16 resistant mutants and their log_2_-transformed MIC values (Figure 2A) and their increases in resistance relative to their parent strains (Figure 2B). Neither correlation was significant (Figure 2), although the former is in the direction one would expect if greater resistance was more costly. However, the latter correlation, which shows no trend at all, is more meaningful because it reflects the relationship between the change in resistance and its associated effect on fitness. In short, we find no support for the hypothesis that the variation in fitness cost among the mutants depends on the level of resistance conferred by their mutations.

### 3.3. Genetic Background Does Not Explain the Variation in Fitness Costs

The 16 ampicillin-resistant mutants evolved from five different parental strains. We tested whether the average cost of resistance varied among the genetic backgrounds or involved an interaction between the backgrounds and level of resistance conferred by the mutations. However, there was no significant effect of either the background (*F*_4,11_ = 1.038, *p* = 0.4310) or interaction (*F*_1,9_ = 0.469, *p* = 0.5110) on the variation in fitness costs among the resistant mutants.

### 3.4. Hitchhiking Does Not Explain the Variation in Fitness Costs

Fourteen of the sixteen ampicillin-resistant clones each have a single mutation, while the Ancestor-2 and Ara−5-1 clones have two and three mutations, respectively [17]. In these latter two cases, one mutation could be responsible for the ampicillin resistance, while the other mutations might be hitchhikers and potentially deleterious with respect to fitness. We therefore compared the average fitness costs of resistant clones with and without secondary mutations. The average cost for the two clones with multiple mutations is higher (23.5%) than for those with a single mutation (8.5%), though the difference is marginally non-significant (Welch’s *t*-test, *t_s_* = 2.1293, 1.6 d.f., one-tailed *p* = 0.0978). However, the small number of cases with multiple mutations limits the power of this comparison. In any case, we find no compelling evidence that hitchhiking of secondary mutations explains the variation in fitness among the resistant mutants.

### 3.5. Genetic Basis for the Idiosyncratic Variation in Fitness Costs

Mutations may have idiosyncratic fitness effects [10]. As a consequence, fitness costs could vary between pathways that confer resistance by different mechanisms, among mutations in different genes within the same pathway, or even between different mutations in the same gene. To explore these possibilities, we used previously obtained genomic data [17] to examine the association, if any, of mutations that arose under ampicillin selection with their corresponding fitness costs.

Three resistant clones have mutations in *ompR* (Ancestor-2, Ancestor-3, and Ara−5-1) and one in *ompF* (Ancestor-4). OmpR is a DNA-binding regulator of the outer-membrane porin OmpF, which allows various solutes to diffuse into the cell and is often implicated in antibiotic resistance [17,23,24]. Despite having mutations in the same regulon, these four clones have variable fitness costs (Figure 1). The *ompR* mutation in the Ara−5-1 clone is associated with one of the highest costs, while the *ompF* mutation in Ancestor-4 is among those without a significant cost. This set of comparisons is complicated, however, by additional mutations in two of the clones with *ompR* mutations: Ancestor-2 has a nonsynonymous mutation in *rpoD* (discussed below), while Ara−5-1 has a large amplification affecting many genes and a presumably neutral synonymous mutation [17]. Further work to make isogenic strains would be required to disentangle which mutations are responsible for the observed fitness differences. The two clones with single mutations in this regulon (Ancestor-3 and Ancestor-4, with mutations in *ompR* and *ompF*, respectively) do not have significantly different fitness costs, given the multiple comparisons (Figure 1), and therefore they do not shed further light on this issue. However, it should be noted that mutations in *ompR* that confer tetracycline resistance do, in fact, vary in their fitness costs in the absence of antibiotic [10].

Three ampicillin-resistant clones (Ara−5-2, Ara+5-1, Ara+5-3) have insertion sequence-mediated deletions that affect multiple genes including *phoE*, which encodes a porin that allows diffusion of phosphate and other small anions into the cell [17]. These three mutants have an average fitness cost of ~26.6%, which puts them among the clones with the highest costs of resistance (Figure 1). Two other clones have point mutations in *rpoB* (Ara+4-1) and *rpoD* (Ancestor-2), which encode the RpoB and RpoD subunits of RNA polymerases, respectively. Both of them are also among those with high costs of resistance (Figure 1), although as discussed above the clone with the *rpoD* mutation also has a mutation in *ompR*.

Summarizing our inferences to this point, 7 of the 16 ampicillin-resistant mutants exhibit fitness costs in the absence of drug (Figure 1). All of those seven have mutations that impact a porin, an RNA polymerase, or both. By contrast, only one of the nine clones without a significant reduction in fitness in the absence of drug has a mutation in those genes or any others that directly impact a porin or polymerase. (The Ancestor-4 clone, with an *ompF* mutation, is the sole exception.) A Fisher’s exact test finds strong support for this putative association between target functions and fitness costs (two-tailed *p* = 0.0014), although it is admittedly a post hoc hypothesis.

The nine clones without significant fitness deficits relative to their sensitive parents all have single mutations [17]. Besides the *ompF* mutation discussed above (Ancestor-4), two of the clones (Ara−6-2, Ara−6-3) have deletions that affect *yfiH*, which encodes a conserved protein of unknown function. Two other clones have amplifications of different genomic regions that affect multiple genes (Ara−6-1, Ara+4-2). Two clones have mutations affecting genes that encode non-global regulatory proteins, *marR* (Ara−5-3) and *slyA* (Ara+4-3). Finally, two clones have mutations in genes that encode proteins involved in synthesis of the cell envelope, *ftsI* (Ancestor-1) and *waaC* (Ara+5-2). In short, several types of mutations that affect many different target genes confer some resistance to ampicillin with minimal or no fitness costs in the drug-free environment used here.

### 3.6. Summary of Results

The ampicillin-resistant mutants in our study grow, on average, about 10% more slowly than their sensitive progenitors in the absence of antibiotics. However, there is substantial variation among the resistant clones in their fitness costs (Figure 1). About half show little or no loss of fitness, while others suffer from deficits of 20% or more. The clones with large deficits have mutations that impact major outer-membrane proteins or RNA polymerases, while the high-fitness clones have mutations in a variety of other genes. The fitness costs appear to be unrelated to the extent of increased resistance conferred by the mutations (Figure 2B). The different genetic backgrounds of the parent strains do not contribute significantly to the fitness costs, nor do secondary mutations that may occasionally hitchhike with mutations that confer resistance. Thus, the striking variation in fitness costs among the ampicillin-resistant clones largely reflects the idiosyncratic effects of the diverse genes and functions affected by the mutations that confer resistance.

## 4. Discussion

In previous work, we investigated how a bacterium’s genetic background affects the evolution of antibiotic resistance, the genetic basis of that resistance, and its associated fitness costs. First, we examined how readily several *E. coli* strains could overcome prior losses of intrinsic resistance when challenged with various antibiotics [16]. We found that resistance potential was more limited in some backgrounds than in others. This result implied that the distinct set of mutations that arose in each population during its history in the drug-free LTEE environment affected its subsequent capacity to evolve resistance. Second, we sequenced the genomes of some of the resistant mutants to assess whether the different founding genotypes took similar or divergent mutational paths to increased resistance [17]. We found that replicate lines evolved from the same genotype tended to have more gene-level mutations in common than those derived from different genotypes. Third, we measured the relative fitness of tetracycline-resistant mutants derived from several parental strains. We asked whether these mutants were less fit than their parents in the absence of antibiotic, and whether the cost of resistance was constant or varied among the mutants [10]. The tetracycline-resistant mutants experienced a reduction in growth rate of ~8%, on average, but with substantial variation in fitness costs. We showed that this heterogeneity reflected, in part, variable costs associated with different mutations in the same target pathway and sometimes even in the same gene.

Here, we extend this work to examine the fitness costs of ampicillin resistance. Ampicillin and tetracycline inhibit cell-wall and protein synthesis, respectively, and resistance mutations in the LTEE-derived lines often occurred in different genes for these two drugs [17]. For example, a large IS*1*-mediated deletion occurred in 3 of the 16 ampicillin-resistant mutants, but in none of the tetracycline-resistant mutants. This deletion affects multiple genes, including *phoE*, which encodes the porin PhoE. However, mutations in *ompR* and *ompF* evolved repeatedly under both ampicillin and tetracycline selection, although they arose more often with tetracycline than with ampicillin (8/16 and 4/16 mutants, respectively) [17]. The *ompF* gene encodes another porin, OmpF, while *ompR* encodes a DNA-binding protein that regulates its expression.

Resistance was often costly in the absence of these drugs. The ampicillin-resistant mutants suffered an average reduction in growth rate of ~10% relative to their sensitive progenitors, and the tetracycline-resistant mutants grew ~8% more slowly [10]. These results are not unexpected because resistance mutations impact cellular physiology and metabolic pathways, and they may also increase the energetic burden on a cell through increased expression of some proteins [4,5,6,7]. While the average reduction in growth rate was large, there was significant variation in the fitness cost among the ampicillin-resistant mutants, as we previously saw for the tetracycline-resistant mutants. As before, we examined several plausible explanations for this heterogeneity.

First, we asked whether mutations that confer greater resistance are more costly than those that confer lesser resistance. If so, then one expects a negative correlation between relative fitness and the level of resistance, either on an absolute basis or, more importantly, relative to the mutants’ progenitors [22]. There was a negative but non-significant correlation with respect to the former, and no trend with respect to the latter (Figure 2). We similarly found no support for this hypothesis in our previous study of tetracycline-resistant mutants [10].

Second, the same or similar resistance mutations might have different fitness costs in different backgrounds [11,12,13,14,15]. For example, Castro and colleagues examined the evolution of resistance to ofloxacin in nine genetically distinct clinical isolates of *Mycobacterium tuberculosis* [15]. They observed significant differences in the frequency of resistance among these strains, and they hypothesized that the differences were driven, in part, by the effect of genetic background on the fitness costs of ofloxacin-resistance mutations. To test this hypothesis, they measured the fitness of each resistant mutant relative to its sensitive counterpart under drug-free conditions. They found that the same *gyrA* mutation had significantly different fitness effects in different genetic backgrounds. In our study, none of the ampicillin-resistant mutants have the exact same point mutation, and we did not construct isogenic strains. However, three mutants from two different genetic backgrounds have identical deletions affecting *phoE* and nearby genes, and they all suffer large fitness costs that are statistically indistinguishable (Figure 1). More broadly, we also tested for trends in average fitness across the five backgrounds in our study. If the background affects the average cost of resistance, then we expect less variation between replicate mutants that evolved from the same parent strain as opposed to different parents. However, genetic background had no appreciable effect on the average fitness cost, and thus it does not explain the variable costs associated with the ampicillin resistance.

Third, individual resistance mutations may have idiosyncratic effects on fitness [10]. The cost of resistance might vary for mutations that impact different physiological pathways, among mutations in different genes within the same pathway, or even between different mutations in the same gene. In our previous work on tetracycline resistance, we found that mutations in different genes within the same pathway and different mutations in the same gene contributed significantly to variation in fitness, even when those mutations occurred in the same genetic background [10]. Specifically, four tetracycline-resistant mutants derived from the LTEE ancestor had significantly different fitness responses, despite conferring similar levels of resistance [16]. One had a mutation in *envZ,* whereas the other three had mutations in *ompR*. These genes encode proteins that comprise a two-component regulatory system often associated with increased antibiotic resistance through altered expression of the porin OmpF [23,24]. Even when we compared two of these ancestor-derived tetracycline-resistant clones, each with a single mutation in *ompR* and no other mutation, the variation in fitness remained significant. By contrast, in the present study of ampicillin-resistant mutants, the variation in fitness costs largely reflects the diverse genes and functions affected by the mutations that confer resistance. All seven ampicillin-resistant clones with large fitness deficits (>20%, on average) have mutations that impact porins (*ompR*, *phoE*), RNA polymerases (*rpoB*, *rpoD*), or both. Only one of the nine clones without a significant fitness cost has a mutation that impacts either of those functions (*ompF*), while the other eight have mutations that affect a variety of different functions. Thus, the mutations that confer resistance to both tetracycline and ampicillin have idiosyncratic effects on fitness. However, the functional level of the idiosyncrasies, or at least our ability to resolve them given the sample sizes, differs between these two antibiotics.

There are many questions about antibiotic resistance that can be examined through the lens of evolutionary biology. Our work here and elsewhere [10,16,17] explores several issues and their intersection. First, how repeatable is the evolution of antibiotic resistance, both phenotypically and genetically, when replicate populations are confronted with the same drug? To what extent does that repeatability depend on genetic background and thus a lineage’s prior evolution? How costly is resistance to the bacteria in the absence of antibiotic? Is the fitness cost the same for all resistant mutants, or does it vary among them? If the cost varies, does it depend on the level of resistance that a mutation confers? Does it depend on the genetic background in which resistance evolved? Or is the cost idiosyncratic, depending on the particular mutation responsible for the resistance?

In our previous work, we first showed that several related *E. coli* strains exhibited subtly different potential for evolving resistance when exposed to various antibiotics [16]. By sequencing the genomes of the mutants and their parents, we showed that the different genetic backgrounds also subtly varied in their tendencies to evolve resistance by different mutational pathways [17]. In the present study, of ampicillin-resistant mutants, and in our previous analysis of tetracycline-resistant mutants [10], we measured the fitness costs of the evolved resistance in the absence of these drugs. In both studies, we found that resistant mutants were, on average, much less fit than their parents. Moreover, in both studies we found the cost of resistance varied significantly among mutants. In neither study, however, was the cost significantly correlated with the level of increased resistance, nor did the cost vary significantly across genetic backgrounds. Instead, in both studies, the cost of resistance was idiosyncratic—that is, it varied depending on the particular mutation—although the details differ between the two antibiotics. For tetracycline resistance, some of the variation in costs resulted from different mutations even in the same target gene [10]; for ampicillin resistance, the variation in costs largely reflects mutations in different sets of genes that were either very costly or nearly cost-free in the absence of drug. The variability in the fitness cost of resistance mutations, as well as the diverse sources of that variation, illustrates some of the complexities associated with antibiotic resistance and underscores the importance of avoiding generalizations when it comes to evolutionary expectations.

## Figures and Tables

**Figure 1 antibiotics-11-00347-f001:**
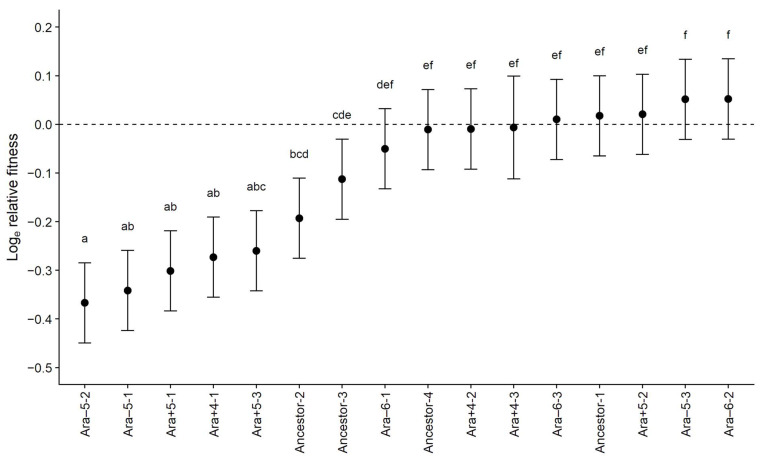
Fitness of 16 ampicillin-resistant mutants, each relative to its parental strain. The mutants are arranged from lowest to highest fitness. Each symbol shows the mean log_e_-transformed fitness based on five-fold replication of paired assays. Error bars show 95% confidence limits calculated using the *t*-distribution with 4 d.f. and the pooled standard deviation from the ANOVA (Table 1). Letters above the error bars identify sets of mutants with relative fitness values that do not differ significantly, based on Tukey’s “honest significant difference” test for multiple comparisons. The dashed line shows the expected relative fitness under the null hypothesis of no cost of resistance.

**Figure 2 antibiotics-11-00347-f002:**
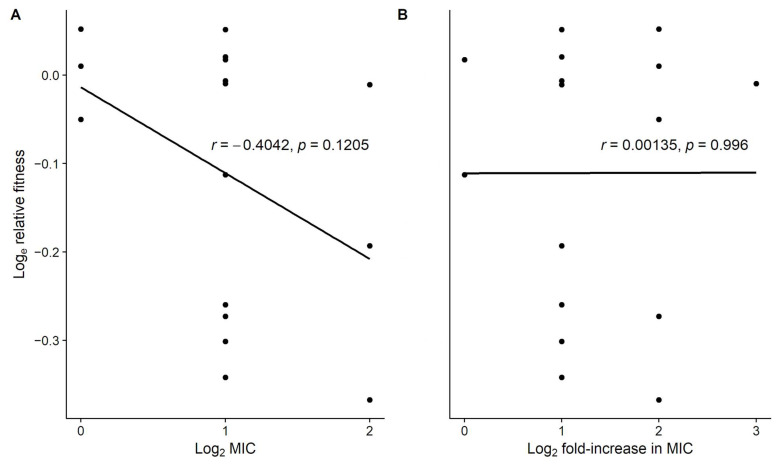
Variation in relative fitness of ampicillin-resistant mutants is not significantly correlated with their resistance level. Correlation between the mean log_e_-transformed fitness of 16 ampicillin-resistant mutants and their (**A**) log_2_-transformed minimum inhibitory concentration (MIC), and (**B**) log_2_-transformed increase in resistance relative to their parental clones after a single round of drug selection [16].

**Table 1 antibiotics-11-00347-t001:** ANOVA on the log_e_-transformed fitness estimates of 16 ampicillin-resistant lines, each measured relative to its sensitive parent.

Source	SS	d.f.	MS	*F*	*p*
Line	1.7220	15	0.1148	26.04	<<0.0001
Error	0.2777	63	0.0044		
Total	1.9997	78			

SS: sum of squares; d.f.: degrees of freedom; MS: mean square; *F*: *F*-ratio; and *p*: *p*-value.

## Data Availability

All data and analysis code for this study are available on GitHub (https://github.com/KyleCard/LTEE-ampicillin-fitness-costs, accessed on 6 February 2022).

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
