# Peer review of "Idiosyncratic Fitness Costs of Ampicillin-Resistant Mutants Derived from a Long-Term Experiment with Escherichia coli"

_antibiotics, 2022, doi:10.3390/antibiotics11030347_

Round 1

Reviewer 1 Report

This study is significant with respect to concerning antibiotic resistance. The introduction is appropriate and the importance of carrying out the study is well explained. It has well designed and procedures well described. Result are presented clearly, with the help of some figures and a table. Discussion is appropriate and correctly referenced, as required, and conclusion supports the hypothesis of the study.

I think they should explain why they use this adjustment model and not another one.

Author Response

We respond (in blue) to each reviewer comment below.

This study is significant with respect to concerning antibiotic resistance. The introduction is appropriate and the importance of carrying out the study is well explained. It has well designed and procedures well described. Result are presented clearly, with the help of some figures and a table. Discussion is appropriate and correctly referenced, as required, and conclusion supports the hypothesis of the study.

Thank you for your positive comments about our study.

I think they should explain why they use this adjustment model and not another one.

We are not sure what the reviewer means by “adjustment model” since that is not a term that we use in our paper, nor have we encountered that term elsewhere. Nonetheless, we tested several different hypotheses regarding the variation in fitness costs among ampicillin-resistant mutants. Specifically, we examined whether level of phenotypic resistance, genetic background, hitchhiking, and/or idiosyncratic effects of specific resistance mutations explains this variation. We explain the rationale behind each hypothesis, and we provide relevant citations and examples in the Discussion section.

Or perhaps the reviewer refers to how we’ve normalized relative fitness values.  In that case, if the fitness of A relative to C (A/C) is x, and the fitness of B relative to C (B/C) is y, then the fitness of A relative to B (A/B) is x/y. We have added a sentence to the Methods that clarifies this point.

Reviewer 2 Report

In this paper, the authors attempt to replicate results they obtained in a previous publication but with another antibiotic. In this publication, they test the effect of evolving to resist ampicillin on growth fitness compared to the parent strain. They show that most of the mutations occur in either porins or RNA polymerases. These mutations often lesser the fitness of the strain compared to the parent strain in regular laboratory conditions, but it does not appear there is a direct correlation between the level of resistance and the level of lost fitness.

This paper is generally well written and I only have a few minor comments:

  • the last paragraph of the introduction (line 78 to 88) looks like a discussion section to me and should be moved or removed.
  • the Davis Mingoli (line 94), Luria Bertani (line 98) and TA agar (line 111) media do not have composition recipes. Though these two media are classic and easy to find, the recipe and source should be indicated.
  • Line 117, "competition assay" is not clear enough. The exact condition of this assay should be further described.
  • The calculations and statistical methods should be described in the methods are they are often confusing for a non-initiate.
  • The legend of table 1 should give the meaning of the abbreviations "SS", "d.f.", "MS", "F" and "p".
  • line 229: what does "IS" stand for?
  • The summary of results (lines 225 to 267) feels unnecessary and does not bring any new information. This should be removed, or placed at the beginning of the discussion.

Author Response

We respond (in blue) to each reviewer comment below.

In this paper, the authors attempt to replicate results they obtained in a previous publication but with another antibiotic. In this publication, they test the effect of evolving to resist ampicillin on growth fitness compared to the parent strain. They show that most of the mutations occur in either porins or RNA polymerases. These mutations often lesser the fitness of the strain compared to the parent strain in regular laboratory conditions, but it does not appear there is a direct correlation between the level of resistance and the level of lost fitness.

This paper is generally well written and I only have a few minor comments:

Thank you for your summary and positive comments.

  • the last paragraph of the introduction (line 78 to 88) looks like a discussion section to me and should be moved or removed.

Thank you for this suggestion. We placed this paragraph at the end of the Introduction section for three reasons. First, it contextualizes the results from the preceding paragraph because we emphasize the value of comparison studies and the important differences between this and our previous work. Second, and related to the first point, it sets the stage for the current work as a follow-up study to Card et al. (2021). Third, this paragraph introduces important implications of our study: namely, that antibiotic resistance is a complex and nuanced problem, and we should therefore avoid generalizations when it comes to evolutionary expectations (a point we reinforce in the Discussion). Given these reasons, and the fact that moving or removing this paragraph will not improve clarity, we have decided not to move or remove it. In the experience of one of our paper’s authors, Richard Lenski, these transitional paragraphs help readers see where the paper is headed—a writing strategy that he learned from a senior colleague decades ago.

  • the Davis Mingoli (line 94), Luria Bertani (line 98) and TA agar (line 111) media do not have composition recipes. Though these two media are classic and easy to find, the recipe and source should be indicated.

We have now provided citations to the media recipes in the Materials and Methods section.

  • Line 117, "competition assay" is not clear enough. The exact condition of this assay should be further described.

Competition assays are used to determine whether one genotype has a fitness advantage over another during head-to-head competition, and to quantify these fitness differences. We discuss the specifics of our competition assays from lines 117 – 131, and in our previous work (Card et al. 2021), which we also cite in this section.

  • The calculations and statistical methods should be described in the methods are they are often confusing for a non-initiate.

We did not provide details of the statistical tests in the Methods section because they are all commonly used approaches that are covered in basic statistics texts (i.e., t-test, ANOVA, linear regression). Instead, we report test parameters, along with each test’s corresponding p-value in parentheses throughout the Results section and in Table 1. Nonetheless, we removed lines 130 and 131 from the Materials and Methods section and replaced them with:

We tested the fitness of each of the 16 ampicillin-resistant mutants and the 16 sensitive parents relative to their respective common competitors with five-fold replication, for a total of 160 competitions. The relative fitness values were loge-transformed before the statistical analyses reported in the Results below. We provide the datasets and analysis code for this study on GitHub at https://github.com/KyleCard/LTEE-ampicillin-fitness-costs.

  • The legend of table 1 should give the meaning of the abbreviations "SS", "d.f.", "MS", "F" and "p".

We have now defined these terms in the Table 1 footer.

  • line 229: what does "IS" stand for?

“IS” refers to “insertion sequence”. We have now edited this line to make this point explicit.

The summary of results (lines 255 to 267) feels unnecessary and does not bring any new information. This should be removed, or placed at the beginning of the discussion.

Although we agree that this paragraph does not add new information, we nevertheless believe that its removal does not improve the paper. As we stated above in regard to your comment about the final paragraph of the Introduction, our experience is that transitional paragraphs of this nature help many readers understand the flow of a paper.

Reviewer 3 Report

The matter of the article is greatly appealing. Bacterial resistances are nowadays of great concern and studies about the fitness of resistant bacteria could be needed to achieve a better understanding about their relevance regarding both clinical aspects and general evolution. Moreover, deeper knowledge about genetic mutations linked to antibiotic exposure and the implications of these mutations on latter resistance to antibiotics could set the basis for better management of antibacterial therapies. Nevertheless, there are some issues to re-consider. The main one is that the text is all the time referring to the tetracycline previous work and it downplays importance to the current study. Certainly, there are no innovative changes comparing that one, except for the antibiotic used. Moreover, there are some points to clarify:

Line 41: You cite the introduction of new drugs as a common practice to regard when combating pathogens. Nevertheless, it is not so ordinary, as not so many new antibiotics are currently developed. Indeed, authorities are claiming for a reduction on their use. For instance, natural origin antimicrobials coming from plants are usually employed in framing for reducing the administration of antibiotics. Revise this statement or clarify it by pointing the kind of drugs you refer to.

Line 46: When talking about mutation, maybe you should consider including an explanation about the selective process involved. Talking about fitness here could be misleading.

Line 70: Why did you use ampicillin for performing this work? I guess that it is because this is the antibiotic used in work 17 for mutant obtaining. However, there are some other penicillins commonly used in medical practice, and maybe a little explanation is needed. In the case of the other work you refer in line 72, tetracycline selection is more understandable because is the “flagship” antibiotic of the family, but why ampicillin for this new paper? Please, argue this issue.

Lines 70-88: You have included a summary of the results at the end of the introduction, but this part should be only in the results-discussion-conclusions section, besides the abstract. Please, clarify the objectives and include them at the end of the section.

Line 71: Clarify the work you are referring to.

Line 78: Which other studies?

Line 87: Which studies? Why have you chosen that protocol?

Line 99: Which concentrations did you selected? Which is the MIC of the strains/clones used? Is the same for all of them? Does it depend on the strain or mutant? Is it related to any specific mutations?

Line 120: Why do you make a 10-fold increase in glucose concentration in the growth media used in the fitness assays compared to the one used for LTEE (lines 94-95).

Line 125: Didn´t you make any intermediate plate-counts?

Line 126: Why did you stopped at day 3? If you base it on a previous work, or even an inner protocol, explain it.

Line 131: Although you refer to a previous work, I consider a deeper explanation about statistics, representations, etc. should be included at the end of Materials and Methods.

Line 180: Figure foots should include information enough to understand the figure. Results might be included in the text.

Line 256: About 10? ~11%

Conclusions section is left. It should be of interest summarizing results section in this part. It will help the readers to make a comprehensive view of the main findings.

¿Have you considered the possibility of the appearance of new mutations during the performance of the fitness assays? I think a new sequencing of the genome would be necessary to assure that the effects observed over bacterial fitness are directly related to prior mutations.

Author Response

We respond (in blue) to each reviewer comment below.

The matter of the article is greatly appealing. Bacterial resistances are nowadays of great concern and studies about the fitness of resistant bacteria could be needed to achieve a better understanding about their relevance regarding both clinical aspects and general evolution. Moreover, deeper knowledge about genetic mutations linked to antibiotic exposure and the implications of these mutations on latter resistance to antibiotics could set the basis for better management of antibacterial therapies.

We agree that this work has potential implications for medicine and public health.

Nevertheless, there are some issues to re-consider. The main one is that the text is all the time referring to the tetracycline previous work and it downplays importance to the current study. Certainly, there are no innovative changes comparing that one, except for the antibiotic used.

Thank you for your feedback. We compare this study to our previous work with tetracycline-resistant lines because we think it is important to highlight whether the same or different factors can explain fitness cost heterogeneity in these otherwise similar cases. This comparison is especially interesting because we used the same experimental conditions, and because the tetracycline- and ampicillin-resistant mutants evolved from the same parental backgrounds.

Moreover, there are some points to clarify:

  • Line 41: You cite the introduction of new drugs as a common practice to regard when combating pathogens. Nevertheless, it is not so ordinary, as not so many new antibiotics are currently developed. Indeed, authorities are claiming for a reduction on their use. For instance, natural origin antimicrobials coming from plants are usually employed in framing for reducing the administration of antibiotics. Revise this statement or clarify it by pointing the kind of drugs you refer to.

Thank you for this suggestion. We agree that this statement could be confusing. Although we meant to refer to the general introduction of new drugs over the past 80 years or so, we have revised this statement by mentioning the discovery of new antimicrobials.   

  • Line 46: When talking about mutation, maybe you should consider including an explanation about the selective process involved. Talking about fitness here could be misleading.

In this statement, we assumed implicitly that resistance mutations are favored in drug environments, but we have now made this point more explicit.

  • Line 70: Why did you use ampicillin for performing this work? I guess that it is because this is the antibiotic used in work 17 for mutant obtaining. However, there are some other penicillins commonly used in medical practice, and maybe a little explanation is needed. In the case of the other work you refer in line 72, tetracycline selection is more understandable because is the “flagship” antibiotic of the family, but why ampicillin for this new paper? Please, argue this issue.

We now briefly address this question in the Materials and Methods, as follows:

We chose ampicillin because it is widely used in both microbiological and evolutionary studies.

  • Lines 70-88: You have included a summary of the results at the end of the introduction, but this part should be only in the results-discussion-conclusions section, besides the abstract. Please, clarify the objectives and include them at the end of the section.

Thank you for this suggestion. However, we disagree that this paragraph only belongs in the Discussion section. As explained in the response to Reviewer #2, it contextualizes our results, frames our study as an important follow-up to our earlier work, and highlights the important implications of our present study as a comparison with our previous study. We argue that it is critical to not “bury the lede” (which means failing to emphasize important parts of a story or report) on these points.

  • Line 71: Clarify the work you are referring to.

We were referring to our earlier work in which we examined the fitness cost of tetracycline resistance (citation #10). Nonetheless, we have now edited this sentence for clarity, as follows:

Here, we extend this work to examine the fitness costs of ampicillin-resistant mutants that evolved from the same parental strains.

  • Line 78: Which other studies?

We are referring to the body of literature, which spans multiple decades and shows that resistance often exacts a fitness cost in the absence of antibiotic pressure (although with some important exceptions, as our own work shows). We cite several of these studies earlier in the Introduction, but we have now added citations #4 - #7 in support of our statement there.

  • Line 87: Which studies? Why have you chosen that protocol?

Please see citation #18 for further information about the LTEE. We do not go into detail about the LTEE protocol because it is not relevant to our study apart from the fact that we subjected the LTEE ancestor and derived clones to antibiotic selection (see citation #16 for further details).

  • Line 99: Which concentrations did you selected? Which is the MIC of the strains/clones used? Is the same for all of them? Does it depend on the strain or mutant? Is it related to any specific mutations?

The MIC values of the resistant strains are variable, as is clearly shown in Figure 2. The effects of specific mutations in relation to their fitness costs (the focus of this paper) are discussed in detail in the Results. See citations #16 and #17 for additional background.

  • Line 120: Why do you make a 10-fold increase in glucose concentration in the growth media used in the fitness assays compared to the one used for LTEE (lines 94-95).

We used a 10-fold higher glucose concentration than in the LTEE because we used that concentration in our earlier study of fitness costs (Card et al. 2021). In that work, we had originally planned to compare fitness measurements based on competitions with measurements obtained from bacterial growth curves. To do so would require a higher population density to make the accurate OD measurements necessary for the growth-curve approach. However, the COVID-19 pandemic prevented us from pursuing this approach. In that study and our current one, we make inferences about the fitness effects of resistance mutations in an arbitrary but well-defined environment, and that environment need not correspond to the particular historical environment.

  • Line 125: Didn´t you make any intermediate plate-counts?

We also assessed the competitors’ frequencies after one day, but we did not use these values to calculate relative fitness. Nonetheless, these data are freely and publicly available on our GitHub repository.

  • Line 126: Why did you stopped at day 3? If you base it on a previous work, or even an inner protocol, explain it.

We stopped our competition experiments at day 3 because, in several cases, mutant frequencies declined to very low levels relative to their sensitive competitors. If we had continued the experiments beyond day 3, these mutant frequencies would have declined to 0, which would have prevented us from quantifying their relative fitness.

  • Line 131: Although you refer to a previous work, I consider a deeper explanation about statistics, representations, etc. should be included at the end of Materials and Methods.

Thank you for this suggestion. As explained in the response to Reviewer #2, we use common statistical methods (i.e., t-test, ANOVA, and linear regression) throughout the manuscript. These basic statistics are not explained in Materials and Methods sections because they are found in countless introductory texts. In any case, we provide the relevant details of our statistical tests in the manuscript. For example, we include details about null hypotheses (lines 137 – 139 and 160 – 161), underlying expectations (lines 142 – 144, 169 – 175, and 307 – 309), and limitations (lines 201 – 202 and line 242 – 244). We have also added further experimental design details to the Materials and Methods section, as outlined above. We should also reiterate that our data are freely and publicly available on our GitHub repository, so anyone can repeat our statistical tests or perform other analyses if they so choose.

  • Line 180: Figure foots should include information enough to understand the figure. Results might be included in the text.

We disagree. In our experience, this information is normally stated as a narrative within the Results section itself.

  • Line 256: About 10? ~11%

Thank you for this correction.

  • Conclusions section is left. It should be of interest summarizing results section in this part. It will help the readers to make a comprehensive view of the main findings.

We believe that the last paragraph of the Discussion section, and the concluding sentence in particular, addresses the reviewer’s concern.

  • ¿Have you considered the possibility of the appearance of new mutations during the performance of the fitness assays? I think a new sequencing of the genome would be necessary to assure that the effects observed over bacterial fitness are directly related to prior mutations.

Although it is possible that mutations arose during our experiment, it is very unlikely that they could reach appreciable frequencies over the three days to alter our results in any systematic and meaningful way. In the LTEE from which our strains are derived, for example, new mutations do not reach appreciable frequencies for about a month. Moreover, such mutations could arise in either competitor, so they would not introduce any directional bias to the fitness estimates.

Round 2

Reviewer 3 Report

Thanks for all your clarifications. Nevertheless, I still wonder the reasons why you have separated both tetracycline and ampicillin resistant studies. You base a great part of the interest of the study in the comparison of the behaviour of both populations coming from a single parental strain. Thus, to ease the reader´s interpretation and increase the impact of the paper,  it would have been preferable to include all the data on a single article.